# An Overview of Chronic Kidney Disease Pathophysiology: The Impact of Gut Dysbiosis and Oral Disease

**DOI:** 10.3390/biomedicines11113033

**Published:** 2023-11-12

**Authors:** Serena Altamura, Davide Pietropaoli, Francesca Lombardi, Rita Del Pinto, Claudio Ferri

**Affiliations:** 1Department of Life, Health & Environmental Sciences, University of L’Aquila, 67100 L’Aquila, Italy; serena.altamura@graduate.univaq.it (S.A.); davide.pietropaoli@univaq.it (D.P.); claudio.ferri@univaq.it (C.F.); 2PhD School in Medicine and Public Health, Center of Oral Diseases, Prevention and Translational Research—Dental Clinic, 67100 L’Aquila, Italy; 3Oral Diseases and Systemic Interactions Study Group (ODISSY Group), 67100 L’Aquila, Italy; 4Center of Oral Diseases, Prevention and Translational Research—Dental Clinic, 67100 L’Aquila, Italy; 5Laboratory of Immunology and Immunopathology, Department of Life, Health & Environmental Sciences, University of L’Aquila, 67100 L’Aquila, Italy; francesca.lombardi@univaq.it; 6Unit of Internal Medicine and Nephrology, Center for Hypertension and Cardiovascular Prevention, San Salvatore Hospital, 67100 L’Aquila, Italy

**Keywords:** chronic kidney disease, inflammation, uremic toxins, gut dysbiosis, oral disease, microbiota, probiotics

## Abstract

Chronic kidney disease (CKD) is a severe condition and a significant public health issue worldwide, carrying the burden of an increased risk of cardiovascular events and mortality. The traditional factors that promote the onset and progression of CKD are cardiometabolic risk factors like hypertension and diabetes, but non-traditional contributors are escalating. Moreover, gut dysbiosis, inflammation, and an impaired immune response are emerging as crucial mechanisms in the disease pathology. The gut microbiome and kidney disease exert a reciprocal influence commonly referred to as “the gut-kidney axis” through the induction of metabolic, immunological, and endocrine alterations. Periodontal diseases are strictly involved in the gut-kidney axis for their impact on the gut microbiota composition and for the metabolic and immunological alterations occurring in and reciprocally affecting both conditions. This review aims to provide an overview of the dynamic biological interconnections between oral health status, gut, and renal pathophysiology, spotlighting the dynamic oral-gut-kidney axis and raising whether periodontal diseases and gut microbiota can be disease modifiers in CKD. By doing so, we try to offer new insights into therapeutic strategies that may enhance the clinical trajectory of CKD patients, ultimately advancing our quest for improved patient outcomes and well-being.

## 1. Introduction

Chronic kidney disease (CKD) is a devastating condition resulting from different disease pathways that irreversibly alter the kidney structure and function over months or years. The diagnosis of CKD is based on evidence of chronic decreased kidney function and an impaired renal structure. The best indicator of overall kidney function is the estimated glomerular filtration rate (eGFR), which represents the total amount of fluid filtered through all functioning nephrons per unit of time [1,2]. Current international guidelines define CKD as a serious condition with a mostly asymptomatic evolution, characterized by diminished kidney function, shown by GFR of less than 60 mL/min per 1.73 m^2^, or markers of kidney damage, i.e., albuminuria (albumin: creatinine ratio ≥ 30 mg/g), or both, lasting at least 3 months, regardless of the underlying cause [3]. To date, around 10–15% of the worldwide population suffers from CKD, with implications on general health [4,5]. The global upsurge in this disease is mainly due to the increase in the prevalence of traditional risk factors associated with its development, including diabetes mellitus, hypertension, and obesity [6,7,8,9,10]. Patients with end-stage renal disease (ESRD), but also patients with slightly reduced kidney function, exhibit a high cardiovascular burden, leading to an increased risk of death, major cardiovascular events, and hospitalization [11,12]. Hereby, a CKD-associated non-traditional risk factor, such as inflammation, is strongly related to a marked increase in cardiovascular mortality. On the other hand, the inability to activate an efficient immune response increases the risk of malignancy and infection [13,14,15,16].

Growing evidence suggests that inflammatory and immune responses can be profoundly influenced by the bidirectional link between the kidneys and other organs, particularly the gut [17]. The relationship between kidney disease and the gut microbiota system is one of the topics of most significant interest, as evidenced by the number of clinical trials recently published or currently underway [18,19,20]. CKD and the alteration (dysbiosis) of the gut microbiota are closely linked in a reciprocal way known as “the gut-kidney axis”, where metabolic and immune pathways are intertwined [21,22,23,24,25]. Of note, recent findings have also highlighted the crucial role of the microbiota-gut-kidney axis in mediating salt-related osmoregulation, allowing small mammals to adapt to high salt loads in a desert habitat [26]. Moreover, the gut microbiota and its interactions with the main components in the brain-gut-kidney axis, such as the neural, hormonal, bone marrow, and immune systems, have been recently described and this network was discussed in the context of CKD and hypertension [22]. Changes in the composition and/or metabolite production of the gut microbiota can influence inflammation, oxidative stress (OS), and fibrosis, thus, offering opportunities to positively manipulate the composition and/or functionality of gut microbiota as a complementary strategy to improve prognosis in renal diseases. CKD and oral diseases also exhibit a multifaceted bidirectional relationship, grounded on shared metabolic and environmental risk factors and encompassing immune responses and mineral metabolism. Compromised immune responses and dysregulated mineral metabolism in CKD patients may indeed exacerbate oral infections and lead to disturbances in tooth and bone homeostasis, respectively, thereby potentially worsening dental and periodontal issues, with a potential impact on gut microbiota.

In this narrative review, we summarize updated literature evidence to provide an overview of the mechanisms linking oral, gut, and renal pathophysiology and raise the question of whether periodontal diseases and gut microbiota can be disease modifiers in CKD.

## 2. Inflammation and Metabolic Products in CKD

Inflammation is a hallmark of deteriorating renal function [27,28,29]. CKD creates a proinflammatory microenvironment caused by infection, uremic milieu, or tissue ischemia [14]. A complex impairment of the immune system occurs in CKD patients, which combines low-grade chronic inflammation and the inability to exhibit protective immune responses. Several clinical entities of kidney diseases and nephropathies induced by hypertension, diabetes, ischemia, or toxic agents lead to sterile inflammation. CKD is characterized by a remarkable increase in proinflammatory cytokine levels, such as tumor necrosis factor-α (TNF-α) and interleukin-6 (IL-6), which are inversely correlated with a decline in eGFR [30,31,32]. IL-6 is considered the most potent inflammatory biomarker for CKD progression, as supported by findings from the multinational STABILITY trial, in which declining eGFR and increasing IL-6 levels predicted incident myocardial infarction, stroke, and all-cause mortality [14]. Notably, in patients with CKD, higher concentrations of high-sensitivity C-reactive protein (hs-CRP), IL-6 and TNF-α and other cytokines, or the expression of interleukin-1α (IL-1α) on the surface of circulating monocytes, are associated with a higher risk of cardiovascular events, cardiovascular mortality, and all-cause mortality, rendering inflammation a ‘non-traditional’ cardiovascular risk factor in CKD [33,34,35].

### 2.1. Inflammasomes in CKD

Inflammation represents a complex network of interactions between renal parenchymal cells and resident immune cells, such as macrophages and dendritic cells, coupled with the recruitment of circulating monocytes, lymphocytes, and neutrophils. Once stimulated, these cells activate specialized structures such as Toll-like receptors (TLRs) and Nod-like receptors (NLRs) that play crucial roles in the recognition of endogenous ligands with associations with a variety of diseases, including CKD [36]. Inflammasomes appear to play an essential role in pathogenetic mechanisms in kidney disease [29]. The NLR family, pyrin domain-containing 3 (NLRP3) inflammasome, is a multiprotein complex comprising NLRP3 receptor protein, apoptosis-associated speck-like protein (ASC), and protease caspase 1 [32]. The inflammasomes can be induced by lipopolysaccharide (LPS) and represent the innate immune signaling pathways that trigger the activation of proinflammatory cytokines in response to various stimuli. The innate immunity system is the first line of host defense that supports homeostasis by regulating endogenous processes such as inflammation and apoptosis. This defense system relies on pattern recognition receptors (PRRs), such as TLR and NLR, that recognize damage-associated molecular patterns (DAMPs) and pathogen-associated molecular patterns (PAMPs) released in response to stress, tissue injury, or apoptosis [37]. By detecting danger-associated molecules, PRRs can set in motion major innate immunity pathways such as nuclear factor ĸB (NF-ĸB) and the NLRP3 inflammasome, causing metabolic reprogramming and phenotype changes of immune and parenchymal cells, triggering the secretion of several inflammatory mediators that can cause irreversible tissue damage and functional loss. Activation of the NLRP3 inflammasome is promoted by TLR activation, thereby activating the NF-κB pathway and the proinflammatory cytokines being released as pro-IL-1*β* and pro-IL-18 [37]. To be converted into their active forms and secreted, the cytokines require caspase cleavage, which causes NLRP3 to oligomerize in the presence of an adaptor molecule ASC. The NLRP3 inflammasome can recruit and activate the proinflammatory protease caspase-1 by recognizing PAMPs or DAMPs [38,39]. PAMPs from invading pathogens, such as LPS, peptidoglycan, muramyl dipeptide, bacterial RNA, and DAMPs from damaged and dying cells in the host, trigger PRRs [40]. PRR binds to the adapter ASC and effector pro-caspase-1 to complete the NLRP3 inflammasome assembly and then promotes pro-caspase-1 self-cleavage to switch on the production of activated caspase-1 [41]. The latter cleaves the precursors of IL-1β and IL-18, producing corresponding mature cytokines, and triggers subsequent inflammatory reactions through proteolysis [42]. In addition to their role in mediating acute kidney disease, the IL-1β/IL-18 axis could also be involved in developing CKD and its complications, accelerated vascular calcification, fibrosis, and sepsis. Vascular inflammation is associated with vascular calcification, and the proinflammatory cytokine IL-18 was the most widely studied component of the NLRP3 inflammasome concerning CKD [43]. The pathophysiology underlying elevated IL-18 levels in CKD may be related to monocyte chemoattractant protein-1 (MCP-1) levels since eGFR was independently associated with serum MCP-1 levels, thus, partially explaining the increased risk of cardiovascular complications in CKD [44]. MCP-1 and ILs are particularly important in CKD progression, by enhancing the activity of adhesion molecules in endothelial cells of renal capillaries [44]. Vascular cell adhesion protein 1 (VCAM-1) and intercellular adhesion molecule 1 (ICAM-1) have been shown to bind to the receptors of activated T lymphocytes, which stimulates fibroblast activity, thus, leading to renal tissue fibrosis and progression of CKD [45].

### 2.2. Oxidative Stress (OS) and Endothelial Dysfunction (ED) in CKD

Inflammation and CKD are associated with increased levels of OS, which promotes immune dysfunction via the activation of several inflammatory signaling pathways [46]. The onset or progression of CKD, hypertension, metabolic syndrome, insulin resistance, and hyperuricemia are closely related to a pro-oxidative state generated by chronic inflammation [47,48]. OS is a pathological condition that can be caused by an impairment of antioxidant mechanisms and an increased production of reactive oxygen species (ROS) [49]. ROS plays a notable role in the physiological regulation of kidney function, rendering the kidney especially vulnerable to redox imbalances and OS. In kidney diseases, ROS overproduction can further enhance the inflammatory response by triggering proinflammatory pathways [50]. Low amounts of pro-oxidative agents, which have essential defensive roles, are usually produced by cells but are inactivated by enzyme systems (e.g., glutathione) and other antioxidants (called scavengers) for their ability to neutralize free radicals. In the kidneys, ROS are mainly produced by the mitochondrial respiratory chain and by enzymes such as nicotinamide adenine dinucleotide phosphate-NADPH-oxidase (NOX) [51]. Some stress stimuli damage mitochondria or impair their function, e.g., NADPH oxidase, inducing ROS production and activating the NLRP3 inflammasome [50]. Active NOX is a dominant source of ROS, and NOX4 is the primary type of NOX in the kidney. Upregulation of NOX4 is crucial in renal OS and kidney injury. Previous studies have demonstrated that intrarenal NOX4 contributes to immune-cell activation in the progression of CKD [52]. NADPH oxidases are involved in ROS production that can lead to proinflammatory signaling through mitogen-activated protein kinases [53]. OS is the major cause of ED. Its presence leads to the oxidization of low-density-lipoprotein-cholesterol (LDL-C) molecules, which, due to atherogenesis, causes damage in arterial intima and inflammatory response in the vessel [54]. In the activated glomerular endothelial cells, inflammation induces the expression of adhesion molecules, E-selectin, and ICAM, which add to renal injury. In CKD patients, elevated levels of inflammatory signals and elevated levels of VCAMs have been associated with higher incidences of sodium and water retention, and disorders in macro-microcirculation [55].

Kidney inflammation is accompanied by endothelial dysfunction and the activation of glomerular and tubular epithelial cells, with the consequent release of inflammatory molecules that, in turn, further recruit immune cells into damaged kidneys. Another recently described mechanism of endothelial dysfunction in CKD is driven by an alteration of the lipid metabolism profile, often conditioned by the coexistence of metabolic disorders, such as diabetes mellitus and obesity [56,57]. Regardless of the cause and degree of renal function deficit, patients with CKD have significant and critical alterations in lipid metabolism [58]. The dyslipidemia observed in this population from the initial stages of the disease and to an increasing extent, the higher proteinuria and the lower GFR, is characterized by quantitative and qualitative changes in lipoproteins, lipolytic enzymes, and lipoprotein receptors, which likely contribute to the progress of this disease [59]. Thus, increased triglyceride levels, diminished high-density lipoprotein-cholesterol (HDL-C), and varying levels of oxidized LDL-C occur as kidney function declines and inflammation becomes more pronounced [60,61]. It is also conceivable that the imbalance of lipid metabolism, which behaves as a pro-atherosclerotic factor, combined with other metabolic disorders, such as obesity and diabetes, contributes significantly to determining the risk of developing cardiovascular disease (CVD), peculiar to patients with CKD [62,63,64].

### 2.3. Uremic Toxins (UTs) and Gut-Derived Uremic Toxins (GDUTs) in CKD

Renal function deteriorates as the disease progresses, especially in case of a lack of proper dietary measures and medical therapy, eventually leading to the retention of uremic toxins (UTs), i.e., solutes usually excreted by the kidneys that impair normal cell physiology [65]. In its final stage, this phenomenon is defined as uremic syndrome [66] and is characterized by a comorbidity profile including mineral-bone disorders, anemia, insulin resistance, sarcopenia, and cognitive impairment [67,68]. In the most advanced stages of the disease, uremic syndrome promotes the reduction of appetite because the products of uremic retention are not eliminated in the urine, which can gradually lead to malnutrition [69]. UTs have been shown to contribute to many uremia-associated dysfunctions, including an altered immune response [70]. Several studies have shown that UTs increase the levels of TNF-α and IL-6 and cause a worsening of the inflammatory state through the promotion of OS [71]. Both inflammation and UTs substantially contribute to the progression of CKD and CKD-associated complications [72]. A CKD-specific risk factor is a group of gut-derived uremic toxins (GDUTs), such as indoxyl sulfate, *p*-cresol, *p*-cresol sulfate, and trimethylamine-*N*-oxide (TMAO). The latter is a derivative of the catabolism of products essentially of animal origin, containing choline, phosphatidylcholine, carnitine, and betaine [73]. *P*-cresol sulfate and indoxyl sulfate, which in CKD patients can reach levels 100 times higher than in healthy subjects, derive instead from the degradation of aromatic amino acids, such as tryptophan, phenylalanine, and tyrosine [74]. These substances have been shown to play a pivotal role in affecting intestinal homeostasis, as well as inducing inflammation and OS in the systemic circulation [75]. Accumulation of toxins and proinflammatory cytokines may constitute DAMPs to which endothelial cells are continuously exposed. In parallel, the increase in toxic substances provokes immunodeficiency in CKD patients, leading to a suppression of humoral and immune responses [76]. Multiple studies demonstrated that the accumulation of GDUTs leads to the occurrence and development of CKD and increases the risk of cardiovascular events in CKD [24,77,78,79,80]. A growing body of evidence focused on the role of UTs in inducing a prothrombotic phenotype characterized by an increased risk of both arterial and venous thrombosis in the CKD population [81,82]. Comorbidities such as diabetes, hypertension, and hypercholesterolemia, which are some of the leading causes of CKD, are also well-known risk factors for thrombosis [83].

In CKD, serum uric acid concentrations increase, contributing to renal tubular damage, endothelial dysfunction, OS, and intrarenal inflammation [84]. Several studies have shown that uric acid can affect the morphology and function of renal parenchymal cells by activating the NLRP3 inflammasome and secreting related inflammatory factors [85,86]. After the cells are stressed, various organelles become dysfunctional and participate in the NLRP3 inflammasome activation process, thereby affecting the occurrence and development of CKD in hyperuricemia [87]. When stimulated by uric acid and uric acid crystals, monocytes and macrophages activate cell membrane TLRs and NLRs that can initiate major innate immune pathways, such as NF-ĸB and NLRP3 inflammasomes [88]. Recent studies on the TLR family have found that TLR2 and TLR4 are closely related to the activation of inflammasomes by uric acid. Uric acid can stimulate TLR4 to increase NF-ĸB transcription and further activate inflammasomes, release IL-1, and increase inflammation [89]. Hyperuricemia is an independent risk factor for the progression of CKD. It is frequently associated with several other conditions, such as hypertension, diabetes, obesity, heart failure, overweight, and CVD [90,91]. High uric acid levels can lead to pathological conditions, such as gout, urinary stones, inflammation, and uric acid nephropathy [92]. Uric acid alarms the innate immune system by activating dendritic cells and macrophages. In the cytosol of dendritic cells and macrophages, the NOD-, leucine-rich repeat-receptor, and NLRP3 inflammasome is assembled and produces IL-1β and IL-18 in response to uric acid. Moreover, uric acid indirectly modulates the adaptive immune system through activated dendritic cells, which manifests as enhanced cytotoxic T cell activity, T helper (Th)2 cell-mediated immune response, and differentiation of naïve T cells toward Th17 cells [48]. Some studies have demonstrated that uric acid directly activates T cells and PI3-kinase (phosphatidylinositol 3-kinase) [93]. Elevated levels of inflammatory cytokines in plasma, and intrarenal infiltration of macrophages and T cells represent the systemic and local nature of chronic inflammation in CKD [94,95]. The pathophysiological mechanisms underlying the inflammatory state in CKD are complex and involve a maladaptive cellular response to injury that leads to persistent activation of proinflammatory and profibrotic signaling [96].

Systemic and chronic proinflammatory states due to CKD contribute to vascular and myocardial remodeling processes resulting in atherosclerotic lesions, vascular calcification, and vascular senescence, as well as myocardial fibrosis and calcification of cardiac valves [97,98]. While numerous risk factors promote atherosclerosis by inducing endothelial dysfunction and its progress to vascular structural damage, CKD affects the medial layer of blood vessels primarily through vascular calcification. Ongoing research has identified vascular calcification as a multifactorial, cell-mediated process in which numerous abnormalities, like mineral dysregulation and especially hyperphosphatemia, induce a phenotype switch of vascular smooth muscle cells to osteoblast-like cells. Hyperphosphatemia, as a frequently observed finding among patients with CKD, is a significant factor involved in the development of medial calcification in CKD [99].

### 2.4. The Bi-Directional Relationship between Visceral Fat and CKD

Adipose tissue is in a continuous state of dynamic remodeling depending on energy reserves and insulin sensitivity. It plays a vital role in establishing homeostasis between energy expenditure and inflammation, as well as thermogenesis, a process that can be significantly dysregulated in CKD. Also, the regulation of adipose tissue-produced adipokines, which are involved in appetite regulation, inflammation, and glucose metabolism, is significantly altered in patients with CKD [100]. Numerous connections exist between obesity and CKD. They share many complex pathophysiological mechanisms (such as hyperinsulinemia, increased OS, chronic inflammation, etc.) as well as several risk factors and associated diseases (e.g., insulin resistance, hypertension, dyslipidemia, endothelial dysfunction, sleep disorders, etc.). In 1974, Weisinger et al. [101] first reported that patients with severe obesity could have large amounts of proteinuria with pathological biopsies, thus, suggesting the presence of kidney damage. Today, overweight or obesity are considered as strong independent and potentially modifiable risk factors for CKD and the development of ESRD. Briefly, growing evidence shows that obesity and kidney disease are closely associated [102,103]. The potential factors linking obesity and CKD include insulin resistance, lipotoxicity, adipokine and cytokine dysregulation, hypertension, and enhanced glomerular blood pressure [58,104,105,106]. Considering that obesity-associated CKD is characterized by proteinuria, glomerulomegaly, progressive glomerulosclerosis, and impaired kidney function [107], the prevention and treatment of obesity-linked CKD are considered very crucial. In the past, weight loss (lifestyle therapies and bariatric surgery) and renin-angiotensin-aldosterone system blockage were the primary therapeutic methods in these patients [108].

Recently, some studies have also focused on the role of genetics in obesity-related CKD. An increased expression of genes has been related to inflammatory factors, insulin resistance, and lipid metabolism in patients with obesity-associated nephropathy [109]. In the future, genetic and metabolomic studies of obese nephropathic patients are needed to provide further options for early diagnosis and treatment.

Because obese individuals often have numerous signs of metabolic syndrome, it is not easy to understand whether the effect of obesity on CKD depends on these metabolic disorders.

Growing evidence suggests that metabolically healthy obesity is associated with a higher risk of CKD events [110,111]. Recent evidence shows that kidney inflammation is crucial in initiating the development and progression of obesity-related CKD [106]. Adipose tissue, particularly visceral adipose tissue, is a primary source of cytokine release in metabolic syndrome, such as leptin, adiponectin, TNF, MCP-1, transforming growth factor (TGF), and angiotensin II (ANG II) [112,113]. TNF-α is one of the most important mediators of adipose tissue inflammation [114]. Notably, TNF-α deficiency protects the kidney from obesity-induced albuminuria and kidney damage [115].

The perirenal fat, the adipose tissue that surrounds the kidneys, other than affecting hemodynamics by exerting local pressure on the kidney, influences renal function through several secretory agents, including cytokines, adipokines, and metabolites, all of which are considered crucial for optimal kidney physiology. Although dyslipidemia is thought to be a consequence of kidney disease, some clinical studies show evidence that altered lipid metabolism may contribute to the pathogenesis and progression of kidney disease [116,117]. Still, more future research is required to confirm the mechanisms by which lipids affect kidney pathology. Perirenal adipose tissue was formerly assumed to provide mainly mechanical kidney support. Nevertheless, investigations have shown that it has a stronger association with kidney illnesses than other visceral fat deposits in obesity or metabolic disorders [118]. Perirenal fat is a metabolically active adipose tissue with endocrine and paracrine roles in glucose and lipid homeostasis and inflammation by generating and secreting adipokines [119]. Increasing evidence suggests that a modification of perirenal fat, also in terms of thickness, is associated with CKD risk and may be used to predict reduced GFR and increased incidence of proteinuria in obese/overweight subjects [120]. Perirenal fat accumulation may directly compress the renal vasculature and parenchyma, leading to increased interstitial hydrostatic pressure, stimulation of renin release, glomerular filtration, and sodium tubular reabsorption, all of which accelerate the progression of renal disease and ultimately lead to decreased GFR [120]. On the contrary, excess perirenal fat can trigger kidney injury through the paracrine or systemic secretion of adipokines and inflammatory factors and activation of the sympathetic and renin-angiotensin-aldosterone systems. For example, leptin, adiponectin, TNF-α, and IL-6 can lead to elevated GFR and increased urinary albumin excretion by altering renal hemodynamics and damaging the vascular endothelium [121,122]. On the other hand, renal dysfunction decisively influences the condition and reactivity of the adipose organ; the latter is significantly altered by kidney disease through multiple mechanisms. Indeed, as previously stated, CKD results in the accumulation of different metabolic waste products or UTs, which can affect both adipocytes and macrophages and promote adipose tissue dysfunction. In macrophages of adipose tissue, UTs also induce excess ROS leading to an inflammatory phenotype. Moreover, adipocyte exposure to UTs results in endoplasmic reticulum stress. ROS facilitates lipolysis, browning, and adipokine dysregulation, leading to insulin resistance and inflammation. There is a crosstalk between adipocytes and macrophages, with adipocyte-derived cytokines facilitating macrophage recruitment to adipose tissue and free fatty acids from lipolysis promoting macrophage metabolic activation with increased inflammatory cytokine production. In addition, macrophages exposed to uremic serum promote inflammatory adipokine production. In summary, adipose tissue metabolic dysregulation in CKD results in increased inflammatory cytokine production, dyslipidemia, and insulin resistance, which promote atherosclerosis, as well as increased thermogenesis and adipocyte catabolism, both associated with cachexia.

In conclusion, perirenal fat is a metabolically and immunologically active tissue vulnerable to inflammatory insults such as cytokines and chemicals and fatty acid overload. In response to these challenges, perirenal fat becomes infiltrated by macrophages and undergoes adipocyte hypertrophy and damage. Adipose inflammation, in turn, elevates circulating inflammatory factors and alters the adipokine profile, thereby contributing to kidney dysfunction. Pharmacological interventions that modulate adipose tissue function have demonstrated kidney-protective effects in models of CKD and diabetic nephropathy [123]. These benefits are likely due, at least in part, to improved adipocyte function, for example, through the increased production of adiponectin with consequent positive effects on renal physiology [123].

## 3. Gut-Kidney Axis and CKD

### 3.1. The Impact of Gut Dysbiosis on CKD

As it has already been widely described by countless studies, in physiological conditions, the gut microbiota influences the well-being of the host by contributing to nutrition, metabolism, and immune function. On the other hand, gut dysbiosis has been implicated in the pathogenesis of diseases such as obesity, type 2 diabetes, inflammatory bowel diseases, CVD, and even neurodegenerative conditions [70,124]. The gut microbiota adaptability is crucial to maintaining gut homeostasis, but drastic alterations due to antibiotics or nutritional factors are potentially harmful [125].

The gut microbiota exhibits essential metabolic functions that can be modified with diet because it uses carbohydrates and proteins in the intestinal lumen [126]. Through saccharolytic fermentation, carbohydrates are converted to short-chain fatty acids (SCFAs), such as acetate, butyrate, and propionate, which possess anti-inflammatory and protective effects on immune function and intestinal barrier integrity tags [127]. In contrast, proteolytic fermentation products, such as phenols, indole, amines, and ammonium, are potentially toxic metabolites and reduce circulating SCFA levels [128].

Renal dysfunction modulates the environment inside the intestinal lumen, which promotes pathogen overgrowth [129,130,131]. Differences in microbial ecosystems have been studied persistently for their involvement in the progression of CKD [132,133]. The altered gut microbiota in CKD patients is characterized by an abundance of Proteobacteria, Enterobacteriaceae, and Clostridium, and a concomitant reduction of Lactobacilli and Bifidobacteria [134]. In this context, gut dysbiosis results in increased proteolytic metabolism and the production of different harmful substances and UTs, able to cause chronic immune activation [135,136]. The gut microbiota interferes with renal function through its metabolites, such as trimethylamine (TMA) and SCFAs, produced by food substrates. For the metabolism of substances such as choline, phosphatidylcholine, and L-carnitine ingested with the diet, the human body does not have an endogenous enzyme kit, and specific intestinal microorganisms transform them into TMA, which is absorbed into the circulation and reaches the liver, where it is oxidized by flavin monooxygenase in TMAO, and finally excreted by the kidneys [137]. For its atherogenic action, TMAO is now recognized as one of the most important independent risk factors for CVD, as well as for the development of cardiovascular complications typical of CKD, and elevated plasma levels of this metabolite have been related to an increase in major cardiac adverse events and mortality in the general population and patients with CKD [72,73,138]. Elevated TMAO concentrations are positively correlated with long-term mortality risk in patients with CKD, atherosclerosis, and heart failure. In two previous studies, higher serum level of TMAO has been regarded as a predictor of coronary atherosclerotic disease and mortality in CKD patients. However, a complete consensus has not yet been reached on this position [139]. Kim et al. reported that the TMAO levels were significantly elevated in CKD, supporting its role as a risk factor in these patients [140]. A recent study showed that TMAO levels were significantly higher in CKD patients than in healthy subjects; in addition, the group with most impaired renal function showed a statistically significant increase in TMAO concentrations compared to the higher GFR group, which in turn had significantly higher TMAO concentrations compared to the control group [141]. Of note, in CKD patients a significant reduction in the intestinal microbiota biodiversity was detected compared to the control group, with a predominance of bacteria belonging to the Proteobacteria group expressing genes involved in TMA metabolism- These results support that the increase in TMAO levels in CKD patients is directly associated to gut dysbiosis [142].

The gut–kidney crosstalk refers to the association between CDK, the gastrointestinal environment, and changes in the intestinal epithelial barrier (IEB) permeability [143,144].

Gut dysbiosis and consequent IEB dysfunction can lead to bacterial translocation, which finally triggers a state of persistent systemic inflammation in CKD patients. When gut dysbiosis occurs, pathogenic bacteria overgrow and secrete increased amounts of bacterial products such as LPS, peptidoglycans, and bacterial DNA and/or outer-membrane proteins into the host circulatory system, causing chronic immune activation. These harmful substances alter intestinal permeability and activate the intestinal-mucosa immune system, thereby promoting the generation of inflammatory factors such as IL-6, interferon-γ (IFN-γ), and TNF-α [17,25,145,146]. Furthermore, neutrophils and monocytes from CKD patients display an exaggerated response to stimulation with LPS, possibly due to the uremic environment that induces an increased expression of TLR2 and TLR4 [31,147].

As pathogen sensors, TLRs, NLRs, and RIG-I-like receptors distributed in the intestinal epithelium can recognize the PAMPs and activate downstream signaling pathways and molecular events able to induce the expression of anti-infective cytokines and other intestinal mucosal immune defense molecules to promote the occurrence of mucosal immune responses, crucial for maintaining gastrointestinal homeostasis [148]. Intestinal intraepithelial lymphocytes and lamina propria T lymphocytes are crucial immune cells of the gut, while dendritic cells are one of the professional antigen-presenting cells. As one of the most important members of the inflammasome family, the NLRP3 inflammasome is widely present in epithelial and immune cells. When the NLRP3 inflammasome is activated by pathogens, such as bacterial toxins and ROS in the gut, its downstream caspase-1 effector proteins activate inflammatory factors, including IL-1β and IL-18, thereby triggering an inflammatory response in the gut [149].

The integrity of the IEB plays a role in regulating host–bacteria homeostasis [150]. Increased gut permeability follows the disruption and depletion of intestinal tight junction structures caused by elevated azotemia, the presence of UTs, and gut dysbiosis, leading to the translocation of both bacteria and endotoxin in patients with CKD [151,152]. Tight junctions are dynamic protein structures localized in the apical portion of enterocytes. They connect both epithelial and endothelial cells. The integrity of tight junctions derives from the integrity of their connections. Since tight junctions control the transition of molecules through the paracellular space, their disruption and depletion lead to the loss of barrier homeostasis and increased intestinal permeability [153].

### 3.2. The Impact of CKD on Gut Microbiota: A Vicious Circle

CKD and the gut microbiota influence each other. Just as gut dysbiosis can affect kidney function, CKD has a substantial impact on the gut microbial profile, which is highly sensitive to the level of UTs [154].

Elevated urea concentrations in body fluids cause it to spread into the intestinal lumen, where it is converted into ammonia by urease-positive species and finally hydrolyzed to ammonium hydroxide [155]. The latter causes the interruption of tight junctions, with damage to the IEB and consequent increased permeability, allowing bacterial toxins to pass into circulation, thus triggering a vicious circle that also aggravates the progression of kidney damage [153,156]. In CKD, the gut microbiota is directly implicated in the increase of UTs due to iatrogenic causes and dysbiosis induced by uremia. In contrast, the progressive impairment of renal function increases the concentration of these toxins in circulation due to reduced renal excretion. The vicious circle worsens CKD and leads to several conditions, including insulin resistance, protein malnutrition, immune dysregulation, and atherosclerosis [57,157,158,159].

UTs promote the imbalance between the formation of ROS and antioxidant capacity, which leads to excessive OS [160]. Increased ROS induce inflammation, endothelial dysfunction, atherosclerosis, and fibrosis and are considered powerful promoters of CKD [77,138,161].

In severe renal failure, the colon becomes the main route of uric acid and oxalate secretion [136]. This condition can account for the expansion of bacterial species that produce uricase. Exposure of intestinal bacteria to urea through gastrointestinal secretions results in the conversion of urea to ammonia via bacterial urease [162]. CKD patients have a greater abundance in urease, uricase, tryptophanase (indole-forming enzyme), and hydroxyphenylacetate decarboxylase (p-cresol-forming enzyme) families, as well as a decrease in those with phosphotransbutyrylase and butyrate kinase (butyrate-forming enzymes) compared to healthy controls [163]. Bacterial urease of the gut microbiota hydrolyzes urea and produces ammonium hydroxide, which raises luminal pH and alters the microbiota composition [162,164]. Ammonium hydroxide itself is caustic and leads to the degradation of tight junction barrier proteins [165]. On the other hand, significant amounts of uric acid and oxalate are secreted into the intestinal lumen in an adaptive response to the decline of their renal excretion [166]. Urea and uric acid released into the gut are alternative substrates for bacterial species that generally use indigestible carbohydrates. All this leads to an alteration of the intestinal bacterial flora and colonization by opportunistic organisms [166]. Thus, the influence of the gut microbiota on the gut–kidney crosstalk plays a fundamental role in CKD, acting reciprocally: on the one hand, CKD significantly modifies the composition and functions of the gut microbiota and contributes to dysbiosis in humans [155]. On the other hand, gut microbiota can manipulate the processes leading to CKD onset and progression through inflammatory, toxic, and neuroendocrine pathways [167].

### 3.3. Unhealthy Eating Habits as Pro-Dysbiosis Factors in CKD

Factors such as increased protein absorption, reduced dietary fiber intake, slower intestinal transit, and frequent oral intake of iron supplements and antibiotics alter the intestinal microbial environment, leading to systemic inflammation and accumulation of UTs [23,166,168]. Patients with CKD, especially those approaching ESRD failure and undergoing maintenance dialysis, frequently experience a gradual decline in their nutritional status [25]. The presence of unhealthy eating habits contributes to frailty in this population, as it leads to metabolic acidosis and nutrition imbalance exhibited as concurrent depletion of body protein and energy reserves, resulting in muscle wasting, sarcopenia, and cachexia [169,170]. In this context, metabolic acidosis plays an important role in accelerated protein catabolism, negative nitrogen balance, and loss of lean body mass in CKD and ESRD [171]. Metabolic acidosis activates proteolysis by activating the ubiquitin-proteasome system and caspase-3 and contributes to insulin resistance and glucocorticoid hypersecretion [172]. Caspase-3 cleaves actomyosin and myofibrils, providing suitable substrates for ubiquitin-proteasome system-mediated degradation [173]. This state of disordered catabolism constitutes a condition known as protein-energy wasting [174]. Data from previous studies showed a reduction in inflammatory biomarkers among patients with different stages of CKD following a diet of non-animal origin [175,176]. High dietary protein intake can cause intraglomerular hypertension, which may result in kidney hyperfiltration, glomerular injury, and proteinuria [177]. A diet with high protein intake might also lead to metabolic acidosis among patients with advanced CKD who have impaired acid excretion and generation of bicarbonate, particularly in the context of protein sourced from animal-based foods [178]. Thus, metabolic acidosis stimulates the production of intra-kidney paracrine hormones, including ANG II, aldosterone, and endothelin 1, that mediate the immediate benefit of increased kidney acid excretion. However, their chronic upregulation promotes inflammation and fibrosis [178]. Dietary acid might also be a risk factor for CKD through intrarenal mechanisms promoting kidney injury and progressive GFR decline [179,180]. Potential mediators of kidney damage from animal protein include dietary acid load, phosphate content, gut dysbiosis, and resultant inflammation. Patients with advanced CKD often develop metabolic acidosis with elevated anion gap due to the accumulation of uremic anions, recently associated with a higher risk of CKD progression and cardiovascular events [181,182]. Reducing the production of UTs, therefore, may be helpful in improving the clinical outcomes of patients with CKD [183].

The protein-bound uremic toxins, such as indoxyl sulfate and p-cresol sulfate, are derived from the byproducts of aromatic amino acid breakdown by the gut microbiota [184,185]. While metabolic acidosis in CKD is aggravated by the high consumption of meat and refined cereals, increasing the dietary acid load, on the other hand, the intake of fruit and vegetables can neutralize the acidosis and its harmful consequences [186,187]. A plant-based diet may reduce GDUTs by increasing fiber intake and modulating the gut microbiota. Dietary fiber, by increasing intestinal motility, decreases the time for fermentation of amino acids, improves the composition of the dysbiotic microbiota, and enhances the excretion of human and bacterial byproducts, thus reducing the formation and/or accumulation of UTs [188]. Conversely, constipation, common among CKD patients, worsens gut dysbiosis, contributes to the uremic status and the risk of hyperkaliemia, and is considered a risk factor for the development and progression of CKD, likely due to the accumulation of UTs and increase of gut dysbiosis [179,184,189]. A healthy intestinal microbiota is essential for the health and well-being of the host. Consistent with this finding, the balance of gut microbiota is well-established to be beneficial to the host and to play a fundamental role in the metabolism of dietary fibers, carbohydrates, and proteins that are not degraded by human enzymes, as well as in vitamins (e.g., B and K) synthesis, and the production of SCFAs, such as acetate, butyrate, and propionate [190]. These compounds can only be produced by the gut microbiota by saccharolytic fermentation of complex carbohydrates, including indigestible dietary fiber. Once absorbed into the circulation, SCFAs are involved in many metabolic pathways at the systemic level, also favorably influencing renal function. SCFAs have antioxidant, immunoregulatory, anti-inflammatory, antihypertensive, and hypoglycemic action, which can counteract pathogenetic mechanisms involved in kidney damage [191,192]. A fiber-rich diet, such as a vegetarian/vegan diet, reduces protein fermentation, increases carbohydrate fermentation, and may improve the dysbiosis associated with CKD by promoting the expansion of saccharolytic bacteria, such as bifidobacteria and lactobacilli, and the reduction in pathogenic bacteria species [193]. A fiber intake also increases the production of SCFA by commensal bacteria that provide energy to the gut microbiota, allowing amino acids that reach the colon to be incorporated into the bacterial proteins and be excreted in feces instead of being fermented to UTs [194]. SCFA also helps to maintain the functionality and integrity of the intestinal barrier, preserves the luminal pH, inhibits the growth of pathogens, and influences intestinal motility [195]. In this sense, supplementation of hemodialysis patients with the SCFA propionate may reduce proinflammatory parameters, OS, and the levels of some GDUTs as well as improve insulin resistance, iron metabolism, and quality of life [196]. 

### 3.4. Can Probiotics Be a Valid Support to Slow the Progression of CKD?

In view of the high correlation between gut dysbiosis and CKD, potential therapies designed to modulate gut microbiota and microbial metabolites could be promising strategies for the prevention of renal disease [197]. In clinical practice, several gut microbiota-targeted therapies have been applied in kidney diseases, including not only dietary intervention but also the administration of probiotics. Probiotics are defined as “live microorganisms that confer a health benefit to the host when administered in adequate concentrations”. Probiotics are made up of bacterial strains, mainly Lactobacillus, Bifidobacterium, and Streptococci. The latter produce bacteriocins, considered the first line of innate defense against infections, as they are able to inhibit the proliferation of pathogenic bacteria, increase the degradation of waste molecules, decrease the inflammatory response, and participate in the immune response, thus restoring the permeability of the intestinal mucosa [198]. In this regard, some studies have evaluated the potential effectiveness of probiotic consumption on renal function in individuals with CKD, demonstrating a significant decrease in plasma levels of azotemia, markers of inflammation, and an increase in serum antioxidants [199,200].

Especially in the advanced phase of CKD, such as ESRD, the reduction of fruit and vegetable intake is indicated to prevent the risk of hyperkalemia and fluid retention. As a result, however, the lack of fiber amplifies the predisposing factors to dysbiosis, such as the slowing of intestinal transit, edema of the intestinal wall, and increased metabolic acidosis [179,187,201]. In addition, ESRD imposes an increase in oral medication intake, such as supplementation of iron and vitamin D analogs, potassium and phosphate chelating agents, and diuretics, inducing proinflammatory gastrointestinal overload [202,203]. For this reason, intestinal metabolism is significantly modified in the uremic population with the prevalence of proteolytic and/or saccharolytic fermentation process and with increased production and reabsorption of intestinal bacterial metabolites [204].

The growing number of experimental and clinical evidence showed that the intake of probiotics promotes beneficial effects on renal parameters, such as improvement of lipid and glycemic profile, attenuation of blood pressure in hypertensive conditions, reduction of uremic toxins, and improvement of CKD condition [205,206,207,208]. The main benefits of consuming probiotics may be related to their anti-inflammatory and antioxidant power, and their properties to modulate gut microbiota [209,210,211,212]. The main strains used for the management of CKD patients belong to the Lactobacillus and Bifidobacterium genera [213,214]. Also, *Akkermansia* has been proposed as a promising potential therapeutic strategy to lower systemic inflammation in CKD patients [135]. In particular, in CKD patients, oral treatment with *Streptococcus thermophilus*, *Lactobacillus acidophilus,* and *Bifidobacterium longum* slowed the progression of CKD [215]. In uremic patients undergoing hemodialysis, oral lactic acid bacteria preparations were able to normalize the composition of the gut microbiota and inhibited the accumulation of UTs, including phenol, p-cresol, and indicant [25]. Oral *Lactobacillus acidophilus* for hemodialysis patients significantly reduced the serum levels of UT dimethylamine [216]. Probiotic supplementation, other than reducing the levels of UTs in blood, also leads to restoring the intestinal microbial balance [21,217]. Of note, the effect is probiotic strain-specific and dependent on the expression of specific functional features [217]. Thus, specific probiotics improve epithelial intestinal integrity thus blocking pathogen entry and adhesion to the IEB [21]. The beneficial effect of Lactobacillus could be attributed to the influence of this probiotic on the permeability and immune status of the intestinal epithelium. Recently, the assumption of *Lactobacillus casei* Zhang has been shown to correct gut dysbiosis, improve renal function, and delay CKD progression by inducing SCFA and nicotinamide level increase in an animal model [218]. In patients with stage 3–5 CKD, oral administration of *L. casei* Zhang also induced a slowdown of the decline of kidney function [218]. Various approaches also based on the use of probiotics, prebiotics, and synbiotics can restore the gut microbiota and intestinal barrier structure and function, thus alleviating CKD and its complications, including CVD and intestinal dysfunction [25].

Although current evidence strongly supports the efficacy of probiotics in relieving or slowing the progression of the disease, further investigations are needed to validate this novel and attractive adjuvant therapeutic approach able to improve outcomes and quality of life of CKD patients [219,220].

## 4. Impaired Oral Health and CKD

The persistent condition of uremia in patients with CKD adversely affects all body systems, including the oral cavity. Recent epidemiological studies suggest that CKD patients appear to be quite susceptible to several oral diseases [221,222]. Renal dysfunction causes an increase in the concentration of urea in serum, as well as in saliva, leading patients, especially those in advanced stages of CKD, to a recurring condition of halitosis due to uremia. In the oral cavity, excess urea is converted into ammonia by the urease–positive oral microbiota, decreasing the salivary flow and leading to dry mouth, better known as xerostomia, that is usually observed in ESRD patients [223,224]. The CKD patient often perceives an unpleasant metallic taste following the ingestion of food, which consequently leads to decreased appetite and contributes to poor nutritional intake and protein-energy wasting syndrome [225,226,227].

In addition to these mechanisms, other factors, including the altered salivary pH, poor oral hygiene, dysbiosis, use of multiple drugs, and alterations in the immune response, can greatly enhance the risk of developing periodontal disease (PD) [228,229].

### 4.1. The Bidirectional Link between CKD and Periodontal Disease (PD)

In this regard, several studies suggested a bidirectional relationship between CKD and PD diseases based on biological hypotheses [230]. On the one hand, malnutrition status, metabolic acidosis, OS, and low-grade inflammation, observed in CKD have repercussions on oral health and periodontium [231]. On the other hand, strong evidence indicated that the presence of PD indirectly affects CKD and consequently contributes to worsening the quality of life [229,232].

A proposed mechanism for the effect of periodontitis on the development of kidney disease is systemic inflammation. Periodontal pathogens have been shown to adhere, invade, and proliferate in coronary endothelial cells leading to atheroma formation and impaired vascular relaxation. CVD and CKD share many risk factors. Thus, it can be assumed that PD exhibits similar effects within the vascular system of the kidney. Both PD and kidney disease are associated with inflammatory markers such as IL-6, IL-8, TNF-α, and IL-1β, and chronic low-level inflammation associated with PD can lead to endothelial dysfunction that plays a role in the pathogenesis of kidney disease in edentulous patients [230]. CKD is strongly linked with endothelial dysfunction [68]. Of note, PD intensifies systemic inflammatory response, enhancing vascular permeability, the expression of endothelial adhesion molecules, such as ICAM-1 and VCAM, and upregulating the TGF-β [233,234,235]. In addition to inflammatory cytokines, endothelial adhesion molecules act directly in vascular injury via the activation of endothelial cells and smooth muscle cell proliferation [236,237,238]. Consequently, endothelial damage will cause renal artery stenosis related to hypertension [239].

The levels of various acute phase proteins in gingival crevicular fluid, such as CRP and pentraxins family protein, are affected by the local periodontitis response. The increase in acute phase protein concentration in plasma promotes the production of proteolytic enzymes in the body, resulting in renal endothelial cell damage, endothelial cell permeability increase, glomerular filtration dysfunction, and ultimately aggravate kidney disease, which is often considered to be an important link between periodontitis and systemic inflammation [240]. Considering these findings, these processes could be associated with proteinuria via glomerular permeability, renal thrombosis, and renal fibrosis, respectively, and result in a deterioration of renal function [239]. An additional inflammatory marker that contributes to the pathogenesis of PD- and CKD-induced lesions is matrix metalloproteinases (MMP). MMPs are a group of enzymes involved in tissue repair and apoptosis, and some of them are upregulated during periodontal inflammation [241]. In the kidneys MMPs are involved in the regulation of inflammatory response, as well as in chronic fibrosis and progression of CKD; thus, PD-induced systemic overexpression of MMPs might contribute to kidney damage [242].

### 4.2. The Association between Oral Dysbiosis and Impaired Kidney Function

Alterations in systemic or local homeostasis triggered by disease in distant organs could be a major factor in altering oral microbiota [236,243]. Of note, the bidirectional relation between CKD and the gut microbiome is well-documented, while changes in the oral ecosystem in CKD are less studied [244,245]. In this regard, biomarker-based human studies have demonstrated that higher IgG levels due to the presence of an elevated periodontal pathogen called “red complex bacteria”, such as *P. gingivalis*, *T. denticola*, *S. noxia, A. actinomycetemcomitans*, and *V. parvula* are linked with impaired kidney function [246,247]. Oral microbiota can directly influence the systemic system through translocation and bacteremia or toxic effects of bacterial compounds, although the most important mechanism seems to be mediated by the immune system and inflammation [248,249,250,251]. On the other hand, increasing evidence supports that oral dysbiosis can significantly alter the composition and function of gut microbiota, with dramatic consequences for the homeostasis of the local and peripheric systems [252].

Regarding direct mechanisms, the “red zone” bacteria of PD can counteract immune response in various ways, prolonging inflammation which can affect multiple organs, including the kidneys [246]. *P. gingivalis,* a keystone subgingival pathogen in provoking PD, harms the local immune system through its ability to evade and impair elements of the host immune-inflammatory system, which alters the growth and development of the entire subgingival biofilm [253,254]. Bacterial adhesion and biofilm formation, closely associated with *P. gingivalis*, may involve the kidney by activating adhesion signals, phagocytosis, inflammatory response, and inducible nitric oxide synthase (iNOS). LPS-mediated activation of the innate immune system might also have systemic consequences relevant to the kidney. Moreover, endothelial cells express TLR2 and TLR4 in a diabetic environment, including in renal microcirculation and translocated *P. gingivalis* from the subgingival biofilm, bind to these cell surface receptors, activating endothelial cells and causing overexpression of adhesion molecules like VCAM-1 and E-selectins [255]. This, in turn, leads to leukocyte margination and glomerular, as well as tubulointerstitial inflammation [256]. LPS-mediated TLR activation through My88 (myeloid differentiation primary response gene 88 (MyD88) normally results in nuclear factor kappa B (NF-kB)-mediated transcription of proinflammatory cytokines, which recruit inflammatory cells of the adaptative immunity and reduce or stop inflammation. However, alternative intracellular cascades such as phosphatidyl inositol 3 kinase (PI3K) or complement factor 5 (C5)-mediated cyclic AMP induction and perhaps other processes that block normal phagolysosome activation of macrophages and neutrophils, cause macrophage immunosuppression and enhanced pathogen survival in vitro and in vivo [229]. Moreover, the continuous increase in cAMP activates PKA in macrophages and destroys the bactericidal function of iNOS, which is dependent on NF-κB [257]. This indicates that phagocytosis by macrophages may induce kidney injury. In addition, the elevation of cAMP leads to a decrease in the production of nitric oxide (NO) by renal endothelial cells. The lack of NO can lead to severe renal fibrosis and enhance the interaction between neutrophils and endothelial cells [258]. Consequently, bacteria contribute to protracting inflammation, and the normally homeostatic host–microbial interactions are changed toward disruptive relationships. On the other hand, CKD-induced modifications of the microenvironment also play a permissive role: both innate and adaptive immunity are impaired. In CKD, dendritic cells and macrophages have a weakened ability to present the antigen, leading to decreased efficiency in monocyte stimulation, impaired phagocytic capabilities of neutrophils, and lessened cytokine secretion [259,260]. In CKD, persistent inflammation leads to a compromised immune system, as is found in PD, characterized by a lessened TLR4 expression, especially in fragile subjects, which are more predisposed to infections [261]. Low TLR4 expression has been associated with reduced synthesis of proinflammatory cytokines in response to LPS challenges [261,262]. It is conceivable that decreased activity or expression of TLR is a main factor for the dysfunction of antigen-presenting cells and predisposition to infection of these patients [261].

Furthermore, OS caused by PD may also adversely affect the kidneys [229]. There are several OS markers, such as malondialdehyde (MDA), 8-hydroxydeoxyguanosine (8-OHdG), and 4-hydroxy-2-nonenal (4-HNE). It is plausible that OS has a significant impact on local periodontal lesions, and the effects of OS on systemic inflammation have been shown by several research groups [263,264]. Moreover, in saliva, MDA and 8-OHdG levels are supposed to be associated with oxidative periodontal lesions, and 4-HNE levels in the saliva may reflect systemic inflammation [264].

PD increases the level of MDA in red blood cells, plasma, and local tissues, which leads to the upregulation of ROS. During periodontitis, ROS and inflammatory cytokines are released from immune cells to eliminate periodontal pathogens [229]. *P. gingivalis* LPS has been detected in various endothelial cells; it triggers the induction of increased ROS, followed by NF-kB-induced inflammation, polynuclear adhesion, and cell apoptosis. ROS produced by leukocytes during inflammation is an important defense mechanism in PD targeting bacterial DNA. However, excessive inflammation triggered by the escape mechanisms described above generates excessive ROS, which leads to a systemic imbalance between prooxidative and antioxidant species with potential effects on various organs, including the kidney [265]. The direct bidirectional mechanisms linking periodontitis and CKD are summarized in Figure 1.

Regarding indirect mechanisms, collectively, the bidirectional crosstalk between oral and gut microbiota develops the well-known oral–gut axis, which plays a crucial role in regulating the pathogenesis of several human diseases [252]. In this context, considering that dysbiosis of oral and gut microbiota highlights the association between PD and CKD and that oral-gut transmission of microbes could represent a further indirect mechanism contributing to CKD, a dynamic and complex oral-gut-kidney axis could be hypothesized. Figure 2 illustrates the plausible indirect linking mechanisms of PD and CKD in the hypothesized oral-gut-kidney axis. The regular monitoring of PD and targeting oral-gut microbial transmission may thus become effective strategies to improve the prevention and treatment of CKD.

## 5. Impact of Periodontal Treatment on Kidney Function

Preserving oral health through PT can counteract systemic inflammatory status associated with CKD, thus restoring kidney function in CKD patients [266]. Indeed, a growing number of studies report a positive effect of periodontal treatment (PT) on renal function in ESRD by improving eGFR and creatinine levels and reducing inflammatory markers, such as IL-6, CRP, and ROS [251,264,267,268,269].

Intensive PT was also associated with improved nutritional parameters and iron availability in patients on peritoneal therapy and hemodialysis [240]. A 3-month-PT significantly reduced the systemic levels of TNF-α and other inflammatory parameters (IL-6, hs-CRP, and pentraxin-3) in patients with CKD [270].

Several groups have evaluated the effects of PT in CKD patients. The improvement in peritoneal dialysis patients’ nutritional and metabolic status has been reported [271]. The standard cycle of PT, consisting of a non-surgical phase and a subsequent surgical phase, was performed. Initial non-surgical treatment included oral hygiene instructions, root planning, polishing, scaling, and curettage. Surgical treatment consisting of gingivectomy or gingivoplasty, conducted six months after the non-surgical phase in patients, as described [271]. In particular, the comparison data before and after completion of PT demonstrated a significant declination of the clinical periodontal status and improved inflammatory markers (hs-CRP). For the nutritional aspect, blood urea nitrogen significantly increased. In addition, the erythropoietin dosage requirement significantly decreased from 8000 to 6000 units/week. In conclusion, PT improved systemic inflammation, nutritional status, and erythropoietin responsiveness in peritoneal dialysis patients with ESRD [271].

A 6-month randomized controlled clinical trial designed to evaluate the effects of non-surgical PT on the clinical response and systemic status of ESRD patients showed significant improvement in periodontal clinical parameters and levels of IL-6, ferritin, albumin, creatinine, blood urea nitrogen, and transferrin [272]. Thus, the authors concluded that non-surgical PT, a relatively simple treatment, may decrease the proinflammatory state in the ESRD population and have beneficial systemic effects further improving the nutritional status.

The Kidney and Periodontal Disease (KAPD) study, a pilot randomized controlled trial on 51 patients, evaluated the effect of a non-surgical PT over 12 months among a high-risk (mostly poor and racial/ethnic minority) population on the kidney and inflammatory biomarker levels [273]. The results showed that biomarkers of endothelial injury and endothelial inflammation declined with PT. However, the lack of a proper control group and complete adherence to intervention limited the ability to detect significant differences.

A nationwide cohort study has reported the effect of non-surgical PT, in particular dental scaling, was significantly associated with lower risks of progression to ESRD, major adverse cardiovascular events, infections, and all-cause mortality in patients with CKD [274]. The authors concluded that their results suggest that dental scaling should be further promoted to improve the clinical outcome of CKD. Randomized controlled trials are warranted to examine the causal relationship of their findings.

Among prophylactic measures, oral hygiene is crucial. Improved oral hygiene has been associated with a decrease in the incidence of CKD [275]. A retrospective longitudinal study reported that the frequency of tooth brushing had a positive impact on eGFR decline or the need for dialysis [276].

A pilot study evaluated whether intensive dental prophylaxis could influence the degree of systemic inflammation indicated by CRP levels. 30 CKD patients aged 6–26 years were examined, 15 receiving intensive prophylaxis and 15 receiving standard treatment (control group) [277]. The authors concluded that intensive dental prophylaxis might be a promising approach to reduce systemic inflammation and subsequently lower premature cardiovascular disease in pediatric patients with CKD, despite the lack of statistical significance, stating also that further research requires a larger patient cohort to enable matched treatment groups with long-term follow-up and molecular detection methods for bacteremia.

In a recent narrative review mainly aimed to present the significant pathophysiologic mechanisms that link CKD and PD, the authors also discuss the effects of PT in CKD patients, concluding that there are still too few prospective studies with endpoints defined according to renal or PD outcomes in these patients at risk [251]. So, although the connection between PD and CKD appears to be well established, as these authors rightly state, PD is not listed as a complication of CKD and is not addressed by current Kidney Disease Improving Global Outcomes guidelines [278]. Additionally, current PD guidelines [279] do not require assessing the presence of CKD in PD patients. Future prospective trials should ascertain if protocolized evaluation and standardized intervention can improve outcomes for these patients, and the evidence on the advantages that PT can provide in terms of improving renal function in patients with CKD is growing, clinical trials are still considered insufficient and, therefore, there are currently no globally implemented guidelines for the treatment of PD in CKD patients.

## 6. Conclusions and Future Perspectives

CKD is a devastating disease characterized by chronic inflammation and is associated with an increased risk of cardiovascular complications. Persistent, low-grade inflammation has been recognized as a critical component of CKD. It plays a unique role in its pathophysiology and is partly responsible for cardiovascular and all-cause mortality in CKD. Several factors contribute to chronic inflammatory status in CKD, including increased production and decreased clearance of pro-inflammatory cytokines, OS and acidosis, chronic and recurrent infections, altered adipose tissue metabolism, and gut dysbiosis. Evidence suggests a bidirectional link between CKD and gut dysbiosis that may contribute to the onset and progression of CKD by mediating increased inflammation and/or generating UTs. Alterations in the composition and/or function of the gut microbiota seem to play an essential role in the pathogenesis of many diseases, such as chronic inflammation, diabetes mellitus, obesity, and CVD. There has been a growing interest in studying the composition of the gut microbiota in patients with CKD and the mechanisms by which gut dysbiosis contributes to CKD progression to identify possible therapeutic targets to improve morbidity and survival in CKD. Recent advances in our understanding of the crucial role of gut microbiota and the pathological consequences of dysbiosis have led to the exploration of various strategies to restore the state of eubiosis. Thus, identifying bacteria associated with chronic systemic inflammation and elucidating the role and mechanisms by which the altered gut microbiota contributes to the inflammatory profile is urgently necessary to improve current CKD therapies. A noteworthy therapeutic approach, widely confirmed by the scientific community, concerns the use of specific probiotics, which, together with proper nutrition, aim to reduce endotoxemia and the concentration of UT in patients with CKD. Therefore, using probiotics to treat intestinal dysbiosis could also positively affect kidney function. Although further studies are needed to elucidate better the molecular relationship between metabolites derived from the gut microbiota, signaling pathways, and CKD, all available evidence indicates their involvement in kidney disease and its progression, opening exciting perspectives for therapeutic interventions targeting gut-kidney axis modulation.

Finally, the bidirectional relationship between CKD and periodontal status has been confirmed by several studies, as it has been pointed out that the reduction of renal function, systemic inflammation, lousy eating habits, and the drug therapy related to it affect the health of periodontal tissues. On the other hand, available data demonstrating the effectiveness of PT on kidney function through improvement of eGFR and reduction in inflammatory markers levels has been amply proven [231,268,272,280]. Improvements in endothelial function have also been documented, which may contribute to increased renal microcirculation and, thus, to a more effective filtration mechanism [281]. In addition, PT is believed to positively influence the progression of kidney disease, cardiovascular risk, and several metabolic disorders conditions, such as diabetes, that further impair renal condition [282,283,284,285].

In conclusion, the results of several studies should alert clinicians and patients to the crucial role of oral health in the control of renal function [286]. There is a need for greater collaboration between dentists and nephrologists. The combination of skills between these two specialists generates enormous advantages regarding quality of care and patient survival. Oral prophylaxis and appropriate dental treatment at an early stage should be intensified in patients with CKD, and periodontal therapy could be incorporated into the treatment planning of CKD patients.

## Figures and Tables

**Figure 1 biomedicines-11-03033-f001:**
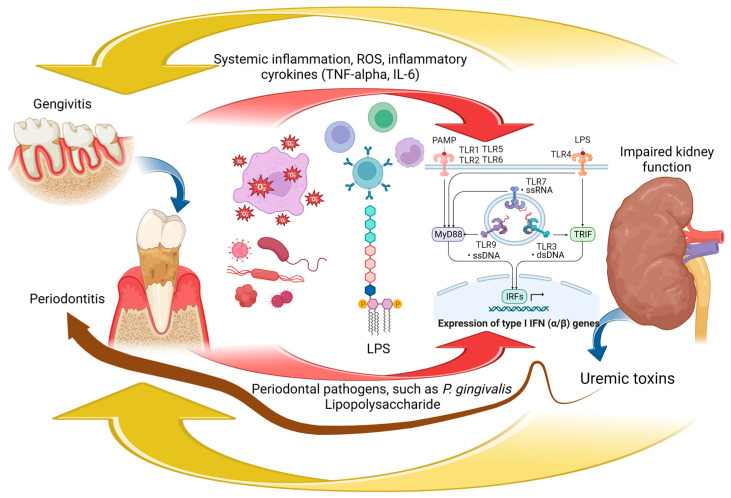
Direct mechanisms linking CKD and periodontitis. Created with BioRender.com, accessed on 11 September 2023.

**Figure 2 biomedicines-11-03033-f002:**
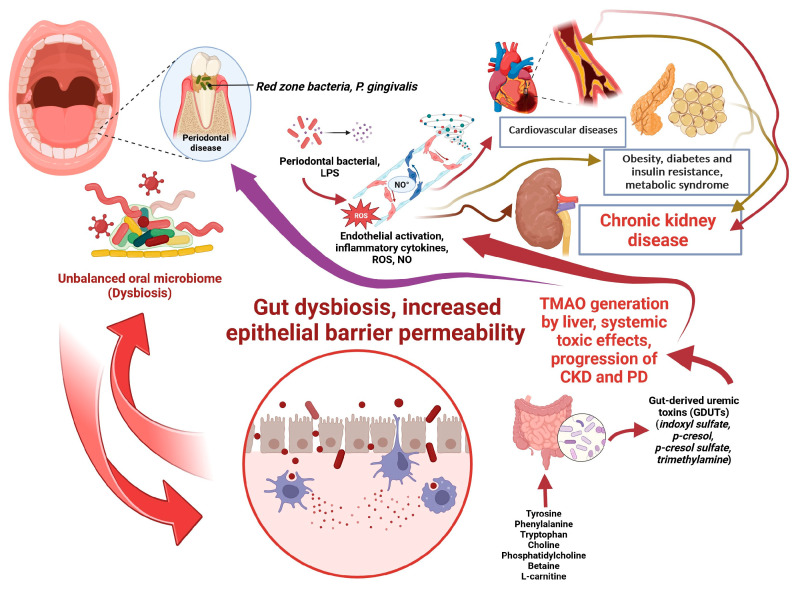
Mechanisms linking PD and CKD in the hypothesized oral-gut-kidney axis. Created with BioRender.com, accessed on 11 September 2023.

## Data Availability

Not applicable.

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
