# Peer review of "An Overview of Chronic Kidney Disease Pathophysiology: The Impact of Gut Dysbiosis and Oral Disease"

_biomedicines, 2023, doi:10.3390/biomedicines11113033_

Round 1

Reviewer 1 Report

Comments and Suggestions for Authors

The manuscript provides a comprehensive exploration of the intricate relationships between oral health, gut microbiota, and kidney function in the context of chronic kidney disease (CKD). It effectively highlights potential clinical implications and probiotic approaches that can benefit CKD patients. Overall, the paper is of good quality and scientifically sound. I have no significant comments; however, there are a couple of matters I would like to recommend to the authors to enhance the paper.

Firstly, the manuscript is quite lengthy, and some sections are dense. To improve readability and flow, it would be beneficial to break down lengthy sections into smaller, more manageable segments. The use of subheadings to delineate different topics would greatly enhance the manuscript's overall organization and make it more reader-friendly. Consider also incorporating additional figure(s) or visual aids to improve reader comprehension.

Secondly, the manuscript's title, "The Oral-Gut-Kidney Axis: Pathophysiology and Therapeutic Implications," implies coverage of both pathophysiology and therapeutic possibilities. However, the authors have included only a brief description of probiotics' role in slowing CKD progression. As a reader, I felt a lack of information on the impact of periodontal treatment on kidney function, which is mentioned in the Conclusions. In my opinion, adding such a section could be a valuable addition to the manuscript. It would help bridge the gap between oral health and kidney function, emphasizing the importance of oral care in managing kidney disease.

Comments on the Quality of English Language

The English quality of the manuscript is generally good in terms of grammar and sentence structure. However, there are some issues with sentence clarity and coherence in certain parts of the text. Some sentences are quite long and complex, which may make it challenging for readers to follow the authors' argument or explanation.

Author Response

We sincerely thank Rev#1 for the helpful comments and criticisms. Accordingly, we have carefully revised and integrated the manuscript. Here is the point-by-point response to your comments and requests.

REV#1

Comments and Suggestions for Authors

The manuscript provides a comprehensive exploration of the intricate relationships between oral health, gut microbiota, and kidney function in the context of chronic kidney disease (CKD). It effectively highlights potential clinical implications and probiotic approaches that can benefit CKD patients. Overall, the paper is of good quality and scientifically sound. I have no significant comments; however, there are a couple of matters I would like to recommend to the authors to enhance the paper.

Firstly, the manuscript is quite lengthy, and some sections are dense. To improve readability and flow, it would be beneficial to break down lengthy sections into smaller, more manageable segments. The use of subheadings to delineate different topics would greatly enhance the manuscript's overall organization and make it more reader-friendly. Consider also incorporating additional figure(s) or visual aids to improve reader comprehension.

Authors’ answer: Many thanks to Rev#1 for the appreciated suggestion. Accordingly, we have split some paragraphs that were quite dense by breaking them down into smaller segments with the following subheadings:

2.1. Inflammasomes in CKD

2.2. Oxidative stress (OS) and endothelial dysfunction (ED) in CKD

2.3. Uremic toxins (UTs) and gut-derived uremic toxins (GDUTs) in CKD

4.1. The bidirectional link between CKD and periodontal disease (PD)

4.2. The association between oral dysbiosis and impaired kidney function

In the new version of the manuscript, we have inserted a new graphical abstract, repurposed Figure 1 in a new light, and introduced a new Figure 2. Also, according to Rev#1's suggestion to divide overly dense paragraphs into smaller sections, the manuscript is now more fluid to read. We prefer not to further burden the review with other schemes on parts considered sufficiently defined by the literature data. Instead, we focused more on the text related to the hypothesized oral-gut-kidney axis and the possible benefits of periodontal therapy in patients with CKD (see below).

Secondly, the manuscript's title, "The Oral-Gut-Kidney Axis: Pathophysiology and Therapeutic Implications," implies coverage of both pathophysiology and therapeutic possibilities. However, the authors have included only a brief description of probiotics' role in slowing CKD progression. As a reader, I felt a lack of information on the impact of periodontal treatment on kidney function, which is mentioned in the Conclusions. In my opinion, adding such a section could be a valuable addition to the manuscript. It would help bridge the gap between oral health and kidney function, emphasizing the importance of oral care in managing kidney disease.

Authors’ answer: Many thanks to Rev#1 for the suggestion that we fully share. The considerations of Rev#1, together with those of the other Revs, made us also reflect on the opportunity to propose a new title to the manuscript, more closely adherent to its content, as follows: An Overview of Chronic Kidney Disease Pathophysiology: The Impact of Gut Dysbiosis and Oral Disease”.

In the revised version of the manuscript, to emphasize the link between oral dysbiosis and CKD, the new subheading 4.1, The bidirectional link between CKD and periodontal disease (PD), has been complemented with other information, considerations, and comments. In particular, we have highlighted the direct and indirect mechanisms through which PD-associated oral dysbiosis could be linked to CKD pathogenesis and progression. In this regard, Figure 1 has been better described, and Figure 2 has been inserted to schematize the plausible indirect mechanisms linking PD and CKD in the hypothesized oral-gut-kidney axis. 

In addition, the description of probiotics' role in slowing CKD progression (3.4. Can probiotics be a valid support to slow the progression of CKD?), rightly considered by Rev#1 to be perhaps too synthetic, has been integrated with additional information to make the context more exhaustive and give further evidence to the role of intestinal dysbiosis in renal disease.

Following the specific request of the Rev#1, a new section 5. Impact of periodontal treatment on kidney function, has been added.

Comments on the Quality of English Language

The English quality of the manuscript is generally good in terms of grammar and sentence structure. However, there are some issues with sentence clarity and coherence in certain parts of the text. Some sentences are quite long and complex, which may make it challenging for readers to follow the authors' argument or explanation.

Authors’ answer: The manuscript was thoroughly revised for style and language.

Reviewer 2 Report

Comments and Suggestions for Authors

The manuscript entitled "The Oral-Gut-Kidney Axis: Pathophysiology and Therapeutic Implications" by Altamura et al. thoroughly examines the biological interconnections among oral, gut, and renal pathophysiology. The authors specifically focus on the intricate oral-gut-kidney axis and explore the potential role of periodontal diseases and the gut microbiota as disease modifiers in chronic kidney disease.

The review article presents a convincing exploration with comprehensive data.

However, to enhance the manuscript's readability, it is recommended to improve the organization of its parts. Particularly, Section 2 is very lengthy. In order to enhance comprehension, it is advisable to organize the content into sub-paragraphs. 

Moreover, it will be useful to summarize key points/mechanisms of each section, from sections 2 to 4, by a figure or a table.

Make more readable the phrase lines: 463-464; 608-609, 616-617

Figure 1 is overly simplistic and more appropriate for a graphical abstract.

Comments on the Quality of English Language

Minor editing of English language required

Author Response

We sincerely thank Rev#2 for the helpful comments and criticisms. Accordingly, we have carefully revised and integrated the manuscript. Here is the point-by-point response to your comments and requests.

REV#2

Comments and Suggestions for Authors

The manuscript entitled "The Oral-Gut-Kidney Axis: Pathophysiology and Therapeutic Implications" by Altamura et al. thoroughly examines the biological interconnections among oral, gut, and renal pathophysiology. The authors specifically focus on the intricate oral-gut-kidney axis and explore the potential role of periodontal diseases and the gut microbiota as disease modifiers in chronic kidney disease.

The review article presents a convincing exploration with comprehensive data.

However, to enhance the manuscript's readability, it is recommended to improve the organization of its parts. Particularly, Section 2 is very lengthy. In order to enhance comprehension, it is advisable to organize the content into sub-paragraphs.

Authors’ answer: Many thanks to Rev#2 for the appreciated suggestion. Accordingly, we have split paragraphs 2 and 4 by breaking them down into smaller segments with the following subheadings:

Paragraph 2

2.1. Inflammasomes in CKD

2.2. Oxidative stress (OS) and endothelial dysfunction (ED) in CKD

2.3. Uremic toxins (UTs) and gut-derived uremic toxins (GDUTs) in CKD

Paragraph 4

4.1. The bidirectional link between CKD and periodontal disease (PD)

4.2. The association between oral dysbiosis and impaired kidney function

Moreover, it will be useful to summarize key points/mechanisms of each section, from sections 2 to 4, by a figure or a table.

Authors’ answer: Thanks to Rev#2 for the suggestion. So, according to Rev#2's suggestion to divide overly dense paragraphs into smaller sections, we think that the manuscript is now more fluid to read. We prefer not to further burden the review with other schemes on parts considered sufficiently defined by the literature data. Instead, we focused more on the text related to the hypothesized oral-gut-kidney axis and the possible benefits of periodontal therapy in patients with CKD. In the new version of the manuscript, we have inserted a new graphical abstract, repurposed Figure 1 in a new light, and introduced a new Figure 2 to schematize the plausible mechanisms linking PD and CKD in the hypothesized oral-gut-kidney axis.

Make more readable the phrase lines: 463-464; 608-609, 616-617

Authors’ answer:

The phrase (lines 463-464): “Consequently, the activation of a chronic local and systemic inflammatory mechanism that unleash induces further damage to the IEB, triggering a vicious circle that also aggravates the progression of kidney damage [152]” has been changed together with the previous sentence to: “The latter causes the interruption of tight junctions, with damage to the IEB and consequent increased permeability, allowing bacterial toxins to pass into circulation, thus triggering a vicious circle that also aggravates the progression of kidney damage [149, 152].

The phrase (lines 608-609): “Oral disorders in these patients can arise either as a result of tissue changes that occur as the disease progresses or as a result of treatments that slow down its course” has been deleted.

The phrase (lines 616-617): “The CKD patient often perceives an unpleasant metallic taste in the mouth following the ingestion of food, which consequently leads to decreased caloric and protein intake, contributing to developing protein-energy wasting syndrome [215-217]” has been changed to “The CKD patient often perceives an unpleasant metallic taste following the ingestion of food, which consequently leads to decreased appetite and contributes to poor nutritional intake and protein-energy wasting syndrome [215-217]”

Figure 1 is overly simplistic and more appropriate for a graphical abstract.

Comments on the Quality of English Language

Minor editing of English language required.

Authors’ answer: The manuscript was thoroughly revised for style and language.

Reviewer 3 Report

Comments and Suggestions for Authors

The review looks fine; however, many parts are quite generalized. 

I would suggest authors read the following article to strengthen their manuscript. To be honest, I didn't find any updates that have not been previously discussed. There are indeed many articles published with more information.  Especially, the Oral-Gut-Kidney Axis section should have more details, figures, and tables.

Nat Rev Nephrol. Author manuscript; available in PMC 2019 Feb 22.   Published in final edited form as: Nat Rev Nephrol. 2018 Jul; 14(7): 442–456. doi: 10.1038/s41581-018-0018-2

Front Med (Lausanne). 2020; 7: 620102. Published online 2021 Jan 21. doi: 10.3389/fmed.2020.620102

Nouri, Z., Zhang, XY., Khakisahneh, S. et al. The microbiota-gut-kidney axis mediates host osmoregulation in a small desert mammal. npj Biofilms Microbiomes 8, 16 (2022). https://doi.org/10.1038/s41522-022-00280-5

Author Response

We sincerely thank Rev#3 for the helpful comments and criticisms. Accordingly, we have carefully revised and integrated the manuscript. Here is the point-by-point response to your comments and requests.

REV#3

Comments and Suggestions for Authors

The review looks fine; however, many parts are quite generalized. 

I would suggest authors read the following article to strengthen their manuscript. To be honest, I didn't find any updates that have not been previously discussed. There are indeed many articles published with more information.  Especially, the Oral-Gut-Kidney Axis section should have more details, figures, and tables.

Nat Rev Nephrol. Author manuscript; available in PMC 2019 Feb 22.   Published in final edited form as: Nat Rev Nephrol. 2018 Jul; 14(7): 442–456. doi: 10.1038/s41581-018-0018-2

Front Med (Lausanne). 2020; 7: 620102. Published online 2021 Jan 21. doi: 10.3389/fmed.2020.620102

Nouri, Z., Zhang, XY., Khakisahneh, S. et al. The microbiota-gut-kidney axis mediates host osmoregulation in a small desert mammal. npj Biofilms Microbiomes 8, 16 (2022). https://doi.org/10.1038/s41522-022-00280-5

Authors’ answer: Many thanks to Rev#3 for the appreciated comments and criticisms. In general terms, the considerations of Rev#3, together with those of the other Revs, made us also reflect on the opportunity to propose a new title to the manuscript, more closely adherent to its content, as follows: An Overview of Chronic Kidney Disease Pathophysiology: The Impact of Gut Dysbiosis and Oral Disease”.

In addition, to enhance the manuscript's readability and to improve the organization of its parts, we have split paragraphs 2 and 4 by breaking them down into smaller segments with the following subheadings:

Paragraph 2

2.1. Inflammasomes in CKD

2.2. Oxidative stress (OS) and endothelial dysfunction (ED) in CKD

2.3. Uremic toxins (UTs) and gut-derived uremic toxins (GDUTs) in CKD

Paragraph 4

4.1. The bidirectional link between CKD and periodontal disease (PD)

4.2. The association between oral dysbiosis and impaired kidney function

Moreover, according to what Rev#3 suggested, in the revised version of the manuscript, to emphasize the link between oral dysbiosis and CKD, the new subheading 4.1, The bidirectional link between CKD and periodontal disease (PD), has been complemented with other information, considerations, and comments. In particular, we have emphasized the direct and indirect mechanisms through which PD-associated oral dysbiosis could be linked to CKD pathogenesis and progression. In this regard, Figure 1 has been better described, and Figure 2 has been introduced to schematize the plausible indirect mechanisms linking PD and CKD in the hypothesized oral-gut-kidney axis. 

 We have also added the new section 5 entitled “Impact of periodontal treatment on kidney function”.

As for the articles recommended by Rev#3, the first one (doi:10.1038/s41581-018-0018-2) was already included in the bibliographic references of the original manuscript (Ref. #22). Anyway, as suggested by Rev#3, the following sentence has been added to highlight the importance of brain-gut-kidney axis in hypertension and CKD:Moreover, the gut microbiota and its interactions with the main components in the brain-gut-kidney axis, such as the neural, hormonal, bone marrow and immune systems, have been recently described and this network discussed in the context of CKD and hypertension [22]”.

The other two articles have been commented on and included in the revised version of the manuscript. In particular,

The second article (doi:10.3389/fmed.2020.620102) recommended by Rev#3 has been cited as ref. n. 21 in the Introduction section.

The third article (doi:10.1038/s41522-022-00280-5) recommended by Rev#3 has been included as ref n. 26 in the Introduction section with the following sentence:

“Of note, recent findings have also highlighted the crucial role of the microbiota-gut-kidney axis in mediating salt-related osmoregulation, allowing small mammals to adapt to high salt loads in a desert habitat [26]”.

Round 2

Reviewer 2 Report

Comments and Suggestions for Authors

The revised manuscript is now suitable for publication.

Reviewer 3 Report

Comments and Suggestions for Authors

The revised version can be accepted for its possible publication.